# Waste Brick Dust as Potential Sorbent of Lead and Cesium from Contaminated Water

**DOI:** 10.3390/ma12101647

**Published:** 2019-05-20

**Authors:** Barbora Doušová, David Koloušek, Miloslav Lhotka, Martin Keppert, Martina Urbanová, Libor Kobera, Jiří Brus

**Affiliations:** 1University of Chemistry and Technology Prague, Faculty of Chemical Technology, Technická 5, 166 28 Praha 6, Czech Republic; koloused@vscht.cz (D.K.); lhotkam@vscht.cz (M.L.); 2Faculty of Civil Engineering, Czech Technical University in Prague, Thákurova 7, 166 29 Praha 6, Czech Republic; martin.keppert@fsv.cvut.cz; 3Institute of Macromolecular Chemistry AS CR, Heyrovského nám. 2, 162 06 Praha 6, Czech Republic; urbanova@imc.cas.cz (M.U.); kobera@imc.cas.cz (L.K.); brus@imc.cas.cz (J.B.)

**Keywords:** waste brick dust, adsorption, lead, cesium, surface complexation, precipitation, solid-state NMR spectroscopy

## Abstract

Adsorption properties of waste brick dust (WBD) were studied by the removing of Pb^II^ and Cs^I^ from an aqueous system. For adsorption experiments, 0.1 M and 0.5 M aqueous solutions of Cs^+^ and Pb^2+^ and two WBD (Libochovice—LB, and Tyn nad Vltavou—TN) in the fraction below 125 µm were used. The structural and surface properties of WBD were characterized by X-ray diffraction (XRD) in combination with solid-state nuclear magnetic resonance (NMR), supplemented by scanning electron microscopy (SEM), specific surface area (S_BET_), total pore volume and zero point of charge (pH_ZPC_). LB was a more amorphous material showing a better adsorption condition than that of TN. The adsorption process indicated better results for Pb^2+^, due to the inner-sphere surface complexation in all Pb^2+^ systems, supported by the formation of insoluble Pb(OH)_2_ precipitation on the sorbent surface. A weak adsorption of Cs^+^ on WBD corresponded to the non-Langmuir adsorption run followed by the outer-sphere surface complexation. The leachability of Pb^2+^ from saturated WBDs varied from 0.001% to 0.3%, while in the case of Cs^+^, 4% to 12% of the initial amount was leached. Both LB and TN met the standards for Pb^II^ adsorption, yet completely failed for any Cs^I^ removal from water systems.

## 1. Introduction

The contamination of waters and soils by toxic carcinogens and radioactive nuclides is a pressing environmental problem. There is particular concern for accumulative poisonous effects in local environments. 

Lead (Pb) is one such element that has well known chronic influence on the central nervous system, where it replaces the residing zinc in neurons. This process can cause mental retardation, behavioral changes, paralysis and anemia. Pb is easily accumulated in body over the lifetime, yet its expulsion is very difficult. It is generally believed that the anthropogenic sources of Pb are limited to the use of tetraethyl lead, however, other Pb origins, such as battery manufacturers, lead smelters and ammunition industries are all major Pb polluters of the environment [1,2]. Cesium as ^137^Cs is an important source of radioactivity from nuclear waste, as it is the major radionuclide of spent nuclear fuel [3]. Due to nuclear accidents over the last decades (e.g., Fukushima in 2011, Chernobyl in 1986), ^137^Cs has infiltrated soils and groundwater and even further into the biosphere thanks to its high solubility [4]. A potential risk of both Pb and ^137^Cs originates from their toxicity and harmful concentrations to organisms in various part of the biosphere [5]. While Pb poisons the soil microorganisms, which limits heterotrophic breakdown of organic matter [6], ^137^Cs is toxic due to its long radio-isotopic half-life (30 years) [7]. Furthermore, ^137^Cs is metabolically similar to potassium, causing it to accumulate in plants and therefore it is incorporated into many food chains [8]. 

Despite a number of modern cleaning technologies (ultrafiltration, reverse osmosis, biological methods, etc.), adsorption remains an effective technique due to its simple application in wastewater treatment. The demand for environmentally and economically friendly technologies has resulted in the development and testing of new natural and synthetic adsorbents, preferably low-cost [5,9,10,11,12]. 

Waste building materials have attained increased professional concern due to their availability and properties analogous to aluminosilicates, such as composition, chemical stability, fine structure and environmentally safe nature. Generally, they are recycled either as concrete admixtures [13] or when fine waste particles have pozzolanic activity, they can be added to cement-based materials to declining the Portland cement consumption [14]. 

Dousova et al. [15] followed out the pilot study on waste brick dusts (WBD) as potential adsorbents of a series of risk cations, anions and radioactive residues. As has been found, WBD showed to be selective cation-active sorbents. The anionic As form (AsO_4_^3−^) was also adsorbed onto WBD, but with more than four times higher sorbent dosage. In terms of possible technological applications, the long-term stability of toxic ions in saturated WBD, as well as its successful incorporation into cement building materials, are crucial.

The aim of this work was to study the characteristics of WBD as a cation-active adsorbent of Pb^II^ and Cs^I^ particles from aqueous solutions. A deep structural characterization of the materials of interest was conducted for potential technological applications based on the combination of X-ray diffraction (XRD) and solid-state NMR spectroscopy; these complementary methods are especially informative. While XRD analysis can easily describe crystalline fractions and changes in mineralogical composition, solid-state NMR spectroscopy gives information about the structure of both amorphous and crystalline phases, local framework defects and extra framework components. Specifically, ^133^Cs and ^207^Pb solid-state NMR spectra can solely provide insight into the chemical nature of adsorbed and solidified contaminants. The information on the structural and surface changes related to the adsorption process may lead to an adsorption mechanism, in which the adsorption rate can be characterized through important parameters such as adsorption capacity, adsorption efficiency, surface complexation and binding energy.

## 2. Materials and Methods 

### 2.1. Waste Brick Dust

Waste brick dusts (WBD) arose as a waste (grinding powder) during the production of vertically perforated ceramic blocks intended for thin joint masonry. The amount of recyclable WBD is limited, as surplus ceramic waste is typically dumped. 

The fraction below 125 µm of two WBD samples from different brick factories in the Czech Republic (Libochovice—LB and Tyn nad Vltavou—TN) were tested as selective cation-active adsorbents for Pb/Cs removal from model aqueous systems. The chemical characteristics of both materials (Table 1) indicates more than 10 times alkalinity, slightly less Fe content and almost double S_BET_ of LB compare to TN. Both WBD were mineralogically similar, with dominant quartz content (ca. 25%), and the presence of illite, albite, anorthite and hematite. The presence of calcite was detected in LB only, while sillimanite was found in TN. 

### 2.2. Model Solutions

Model solutions of Pb^2+^ and Cs^+^ were prepared from inorganic salts (PbCl_2_ and CsCl, respectively) of analytical grade and distilled water, with concentrations of 0.1 and 0.5 mM L^−1^ and their natural pH values (≈3.5). The concentration ranges were selected as appropriate to simulate a slightly increased amount of the contaminant in a water system to a heavily contaminated solution. 

### 2.3. Sorption Experiments

The suspension of model solution (50 mL) and defined dosage (1–15 g L^−1^ for Pb and 1–30 g L^−1^ for Cs) of WBD was agitated in a batch manner at laboratory temperature (20 °C) for 24 h [16]. The product was filtered, and the filtrate was analyzed for residual concentration of cations, while the solid residue (saturated WBD) was kept for subsequent solid-state analyses (NMR, SEM, S_BET_). 

All adsorption data were fitted to the Langmuir isotherm [17,18], according to the equations (1) and (2), which was verified in many papers [9,15,16,17] as a suitable adsorption model for natural sorbents, including aluminosilicates and soils. The obvious adsorption parameters (*q*_max_—maximum equilibrium adsorption capacity; *Q*_t_—theoretical adsorption capacity; *R*^2^—correlation factor; *K*_L_—Langmuir adsorption constant) were then used to evaluate the effectiveness of adsorption systems.
(1)$$q=\frac{Q_{t}\cdot K_{L}\cdot c}{1+K_{L}\cdot c},$$
and its linearized form:(2)$$\frac{1}{q}=\frac{1}{Q_{t}}+\frac{1}{Q_{t}\cdot K_{L}\cdot c},$$
where *q* is an equilibrium concentration of adsorbed ion (adsorbate) in solid phase [mol g^−1^], *c* is an equilibrium concentration of adsorbate in solution [mol L^−1^], *Q*_t_ indicates the theoretical adsorption capacity [mol g^−1^], and *K*_L_ is a Langmuir adsorption constant [L mol^−1^].

The equilibrium amount (*q*) of adsorbate caught in the solid phase was calculated from experimental data by Equation (3):(3)$$q=\frac{V_{0}(c_{0}-c)}{m},$$
where *V*_0_ is the volume of solution [L], *c*_0_ is the initial concentration of adsorbate in solution [mol L^−1^], and *m* is the mass of solid phase [g]. 

### 2.4. Leaching Test

The stability of cations in the saturated WBDs was tested by a standard procedure EN 12457–2 [19]. First, a suspension of dry sorbent and distilled H_2_O at a ratio of 1:10 was agitated in a sealed polyethylene bottle at 20 °C for 24 h. The suspension was filtered out and the filtrate was analyzed for residual Pb^2+^/Cs^+^ concentration.

### 2.5. Analytical Methods

Powder XRD was measured with a Seifert XRD 3000P diffractometer (Seifert, Ahrensburg, Germany) with CoK_α_ radiation (λ= 0.179026 nm, graphite monochromator, goniometer with Bragg–Brentano geometry) in the 2θ range of 5–60° with a step size of 0.05° 2θ. 

The X-ray Fluorescence (XRF) analyses of the solid phase were carried out using an ARL 9400 XP+ spectrometer (ARL, Ecublens, Switzerland) at a voltage of 20–60 kV, probe current of 40–80 mA, and with an effective area of 490.6 mm^2^. UniQuant 4 software was used to evaluate the data (Thermo ARL, Ecublens, Switzerland).

The S_BET_ was measured on a Micromeritics ASAP 2020 (accelerated surface area and porosimetry) analyzer using gas sorption. The ASAP 2020 model (Micromeritics, Norcross, GA, USA) assesses single and multipoint BET surface area, Langmuir surface area, Temkin and Freundlich isotherm analysis, pore volume and pore area distributions in the micro- and macro-pore ranges. The macro-pore and micro-pore samples were analyzed by the Horvath-Kavazoe method (BJH method), respectively. The BHJ method used N_2_ as the analysis adsorptive and an analysis bath temperature of −195.8 °C. Samples were degassed at 313 K for 1000 min.

Solid-state NMR spectra were measured at 11.7 T using a Bruker AVANCE III HD 500 WB/US NMR spectrometer (Bruker, Karlsruhe, Germany). The ^27^Al Magic Angle Spinning Nuclear Magnetic Resonance Spectroscopy (MAS NMR)spectra were acquired at a spinning frequency of 11 kHz, Larmor frequency of 130.287 MHz and recycle delay of 2 s. The spectra were referenced to the external standard Al(NO_3_)_3_ (0 ppm). The ^29^Si MAS NMR spectra were acquired at a spinning frequency of 11 kHz, Larmor frequency of 99.325 MHz and recycle delay of 10 s. The number of scans for the acquisition of a single ^29^Si MAS NMR spectrum was 6144. The spectra were referenced to the external standard M_8_Q_8_ (−109.8 ppm). ^133^Cs MAS NMR spectra were acquired at a spinning frequency of 11 kHz, Larmor frequency of 65.601 MHz, recycle delay of 2 s and the number of scans of 73,000. The spectra were referenced to the external standard CsOH·H_2_O (124.1 ppm). Due to the large chemical shift anisotropy the ^207^Pb NMR spectra were measured using the ^207^Pb MAS NMR and WURST-QCPMG NMR experiments combined with variable offset cumulative spectroscopy (VOCS) technique [20]. The Larmor frequency was 104.640 MHz and the recycle delay was 2 s for all ^207^Pb NMR spectra. The number of scans was set to 30000 for each sub-spectrum. The applied experimental parameters were optimized using a mixture of PbO and Pb(OH)_2_ the resonance frequency. The ^207^Pb NMR spectra were referenced to the external standard Pb(NO_3_)_2_ (−3473.6 ppm). High-power ^1^H decoupling (SPINAL64 for MAS experiments and Continuous Wave (CW) for static experiments) was used to eliminate heteronuclear dipolar couplings in all measurements. The NMR experiments were performed at a temperature of 303 K, and temperature calibration was done to compensate for the frictional heating of the samples [21]. 

The structure of samples was determined using scanning electron microscopy (SEM) on the Tescan Vega 3 (Brno-Kohoutovice, Czech Republic). Energy dispersion spectrometry (EDS) was conducted on the Inca 350 spectrometer (Oxford Instruments, Abingdon, UK).

The concentration of Pb and Cs in aqueous solutions was measured by atomic absorption spectrometry (AAS) using a SpectrAA-880 VGA 77 unit (Varian, Palo Alto, CA, USA) in flame mode. An accuracy of AAS analyses was guaranteed by the Laboratory of Atomic Absorption Spectrometry of UCT Prague, CR, with the detection limit of 0.5 μg L^−1^, with a standard deviation ranging from 5%–10% of the mean).

## 3. Results and Discussion

### 3.1. Specification of WBD Adsorbents in Relation to Pb/Cs Chemistry and pH Value

The chemical and mineralogical composition of both WBD are similar apart from the high content of alkalis in LB as compared to TN, which results in the former exhibiting a significant higher pH (see Table 1). Whereas the pH/pH_ZPC_ values (Table 1) are responsible for the selectivity of adsorption, the points of pH_ZPC_ in relation with pH ranges of adsorption solution are shown in Figure 1. At the pH_ZPC_ of LB (the crosses in both phase diagrams) lower than the pH range of adsorption solution (the dashed areas in the phase diagrams), when pH/pH_ZPC_ > 2, the sorbent surface primarily deprotonates to balance the pH gradient and the negatively charged surface of LB then attracts cations from the solution [22]. In the case of TN, the negligible pH/pH_ZPC_ gradient (pH/pH_ZPC_ ≈ 1, spotted areas vs. the empty rings in the phase diagrams) caused a weak attraction at the solid-liquid interface. In terms of WBD with the average pH_ZPC_ of 5, a higher pH of adsorption has been essential for selective removal of Pb^2+^/Cs^+^. The optimal pH resulting from adsorption experiments ranged from 7.5 to 10.

As shown in Table 1, the S_BET_ and total porosity values also favored the adsorption selectivity of LB to TN, because a more porous material with a larger surface area would generally provide more sorption sites for surface binding.

The adsorption behavior of the investigated systems was also based on the different aqueous chemistry of Pb and Cs, respectively (Table 2, Figure 1). In an aqueous environment, Cs^+^ was expected to form a small hydrated ion, because of its low charge and large crystallographic radius (1.69 Å). While the adsorption of Cs was minimally pH dependent, Cs^+^ ions were bound primarily to cation exchange sites, forming only outer-sphere complexes. On the other hand, Pb^2+^ (1.32 Å) was subjected to high hydrolysis, easily losing the part of its primary hydration shell, especially at high pH [23]. The hydrolyzed ions initially formed strong inner-sphere complexes with sorbent surfaces and eventually led to the formation of polynuclear complexes or surface precipitates [24].

### 3.2. Structural Changes Following Adsorption 

The SEM micro-observation of initial and Cs^+^/Pb^2+^ saturated WBD (Figure 2) showed completely different morphology of surface particles, confirming the above described unequal hydrolyses and binding strategies of Cs^+^ and Pb^2+^ ions. Separately hydrated Cs^+^ ions, that formed outer-sphere complexes with active surface sites, did not change the surface morphology remarkably (Figure 2), while the predicted formation of surface precipitates of Pb_4_(OH)_4_^4+^/Pb_6_(OH)_8_^4+^ (Figure 1) led to the homogeneous covering of sorbent surface. 

The MAS NMR spectra of both materials recorded before and after Cs^+^/Pb^2+^adsorption indicate structural changes, crystallinity and framework defects resulting from the adsorption process. The ^27^Al MAS NMR spectra (Figure 3) demonstrate that aluminum sites in both systems occupy tetrahedral coordination Al^IV^, as reflected by the most intense signals at 60 and 58 ppm for LB and TN systems, respectively. The low-intensity signal at approximately 5 ppm reflects a small amount of hexagonally coordinated aluminum species present in the LB system. Additional information relating to the crystallinity of investigated systems can be deduced from the line-widths of the recorded signals. While the LB system is represented by a relatively broad signal with the half-width of about 1.6 kHz, the TN system is reflected by considerably narrower signal (1.3 kHz). This finding, which agrees with the ^29^Si MAS NMR spectra (discussed later), indicates a more disordered and amorphous character of aluminosilicate framework of the LB system. 

The ^29^Si MAS NMR spectra (Figure 4) provide more detailed insight into the framework of investigated systems and further highlight their distinct structures. The main signal at −91 ppm is clearly attributed to the Q^3^(0Al) sites in dehydroxylated illite, whereas the shoulder at −85 ppm is assigned to a mixture of Q^3^(1Al) and Q^3^(2Al) usually present in natural illite and related systems. The signal resonating at approximately -75 ppm likely reflects the incompletely condensed Q^1^ and Q^2^ units of mineral defects and the amorphous SiO_2_ fraction. The narrow signal at −108 ppm therefore reflects the Q^4^ units of crystalline SiO_2_ (quartz, approx. 20%). 

On the other hand, the ^29^Si MAS NMR spectrum of neat brick dust from TN reflects a much narrower distribution of aluminosilicate building blocks. The signal is dominated by the narrow resonance at −108 ppm, reflecting the Q^4^ units of crystalline SiO_2_ (quartz, approx. 70%). Deconvolution of the recorded spectrum further revealed additional much broader resonance centered at −92 ppm, which would reflect the Q^3^(0Al) units in dehydroxylated illite (approx. 30%). 

The structural changes associated with the sorption process were mostly observed at the ^29^Si MAS NMR spectra of LB system (Figure 4). The intensities of broad signals of Q^1^, Q^2^ and Q^3^ units considerably decreased relative to the signal intensity of Q^4^ units of crystalline SiO_2_. In the case of Pb^2+^ adsorption, the observed decrease is more intense in comparison with the decrease observed during the sorption of Cs^+^ ions, which indicates the partial dissolution of disordered and amorphous fractions rich in the SiO_2_ phase. In contrast, there were no considerable changes in the ^29^Si MAS NMR spectra of the TN system, thus exhibiting structural stability of this material during the adsorption of Pb^2+^ and Cs^+^ ions.

### 3.3. Adsorption of Pb^2+^/Cs^+^ on WBD

The adsorption series were performed under the same conditions (described in Section 2.3) in order to evaluate the applicability of WBD for the removal of toxic cations from aqueous systems. 

The Langmuir adsorption isotherms (Figure 5) illustrate improved adsorption properties of LB for both cations and a considerably higher affinity of Pb^2+^ for both WBD. As shown by the adsorption parameters calculated from the linearized Langmuir equation (Section 2.3, Table 3), only Pb^2+^ adsorption corresponded to the Langmuir model indicating an inner-sphere surface complexation [26,27] for all Pb^2+^ adsorptions. The differences in theoretical adsorption capacities (*Q*_theor_) and binding energies (*K*_L_) of LB/Pb^2+^ as compared to the TN/Pb^2+^ systems resulted from the different pH/pH_ZPC_ ratios (Section 3.1), which could be considered the prior adsorption power [15,16]. Accordingly, LB/Pb^2+^ adsorption system showed much more robust process in all aspects (*Q*_theor_, *K*_L_, R^2^) (Table 3). 

On the other hand, the adsorption of hydrated Cs^+^ ions did not follow Langmuir run, but typical physical adsorption with a weak electrostatic surface binding (outer-sphere complexation) was observed. Maximum equilibrium sorption capacities (*q*_max_) of both LB and TN for Cs^+^ were one to three orders of magnitude lower than for Pb^2+^. 

An almost 100% adsorption efficiency at low sorbent consumption for Pb^2+^ (Figure 6) supported the abovementioned formation of poorly soluble Pb(OH)_2_ clusters on the sorbent surface, which markedly increased the adsorption yields.

The fate of adsorbed Cs^+^/Pb^2+^ ions was also observed in the ^113^Cs and ^207^Pb solid-state NMR spectra. The spectra were recorded only for the LB-system, which in both cases exhibited considerably higher sorption capacities. The ^133^Cs MAS NMR spectrum (Figure 7) shows relatively broad signal covering the frequency range from −7 to approx. −20 ppm. According to literature data, these resonances can be assigned to surface species, in which more shielded (more negative) resonances reflect Cs^+^ sites more tightly bound to the surface, whereas the less shielded resonances correspond to less tightly bound species. This phenomenon can be attributed to the formation of multilayer deposits of Cs^+^ ions.

When measuring ^207^Pb solid-state NMR spectra, there is relatively low natural abundance of ^207^Pb spins (approx. 22.6 %, I=1/2), which is combined with usually very large chemical shift anisotropy (CSA). Consequently, under the standard spinning speeds (10–30 kHz) manifolds of the central signals and spinning sidebands in ^207^Pb MAS NMR spectra can cover extremely broad spectral windows 2000–4000 ppm, rendering them nearly undetectable. Therefore, the attempt here to record ^207^Pb solid-state NMR spectra was nearly unsuccessful. This effect was observed even in the model system, which consisted of a mixture of neat powdered PbO and Pb(OH)_2_ (Figure 8a). However, the application of an alternative experimental technique, ^207^Pb Wideband Uniform-Rate Smooth Truncation-Quadrupolar Carr-Purcell Meiboom-Gill (WURST-QCPMG) NMR, seems to be more suitable for the measurement of broad spectral lines. As demonstrated in Figure 8b, the intensities of the recorded ^207^Pb WURST-QCPMG NMR signals are considerably higher than that recorded by the standard ^207^Pb MAS NMR experiment.

In spite of a long accumulation time (48 h), the resulting spectrum is very poor, indicating very low amounts of the surface Pb^2+^ species (Figure 8c). Nevertheless, two spectral regions with ^207^Pb NMR resonances were identified in the recorded spectrum. One of them covers frequencies from 1200 to 2200 ppm, whereas the second one ranges from −400 to −1000 ppm. These results imply that Pb^2+^ ions on the surface of LB-system form two chemically distinct fractions. One of these fractions is likely structurally related to precipitated hydroxide species, whereas the second one can be rather attributed to Pb^2+^ binding to a silicate tetrahedral [28]. 

Consistent with the previously mentioned results, the adsorption of Cs^+^ on WBDs was worse in all aspects when compared to Pb^2+^. A low effect of Cs^+^ adsorption was mainly given by steric properties. According to Rajec et al. [29], the majority of active sorption sites for Cs^+^ binding have been associated with permanent (pH independent) charge cation exchange sites on zeolites and smectite clays rather than with surface hydroxyl sites on quartz and feldspars. Therefore, zeolites have been frequently used to remove radionuclides and metal cations because of their molecular sieve structure and high cation exchange capacity [30]. Contrastingly, WBDs consist mostly of quartz, feldspars and hematite and exhibit rather poor Cs adsorbing behavior [31]. 

### 3.4. Leachability of Pb^2+^/Cs^+^ from Saturated Sorbents

The stability of cations in saturated WBD was tested by the leaching test (Section 2.4). Not surprisingly, the stability of Pb^2+^/Cs^+^ in saturated WBD was closely related to the selectivity of a particular adsorption (Table 4); Pb^2+^ adsorption on LB carried out by high sorption capacities and efficiencies, linking the formation of very stable inner-sphere surface complexes at both initial concentrations (only about 0.001% of Pb^2+^ was leached). The favorable stability (less than 0.3% of the original amount leached) was recorded for Pb^2+^ in TN. The non-selective Cs^+^ adsorption resulted in a low binding stability during the leaching test, where 4 to 12% of the initial amount was leached. According to the standards (CSN EN 12457), LB and TN saturated with Pb^2+^ fall into the non-hazardous waste. Although the regulation has not specified the limits for a Cs hazard, a high leachability of Cs^+^ from both saturated WBDs (≈ 10% wt.) indicates its environmental risk. In line with the previous results, LB proved better adsorption selectivity and leaching stability to cations than TN. The stability of testified adsorption systems decreased in the following order: Pb^2+^/LB >> Pb^2+^/TN >> Cs^+^/LB > Cs^+^/TN. 

## 4. Conclusions

The adsorption selectivity of WDBs to Pb(II) and Cs(I) was initially affected by the structural properties and morphology of WBDs being tested as Pb^2+^/Cs^+^ adsorbents. The unequal binding forms of both cations presented the next important aspect of adsorption process; while Cs^+^ ions formed separate particles weakly bound by outer-sphere surface complexes, Pb^2+^ appeared in two co-existed phases, namely as the hydrolyzed particles Pb^2+^ bound by strong inner-sphere complexes, and the poorly soluble Pb(OH)_2_ clusters that catch on the sorbent surface. Therefore, the adsorption of Pb^2+^ was much more effective, strongly supported by the almost insoluble surface precipitation (Pb(OH)_2_). In connection with that process, the adsorption of Pb^2+^ was pH dependent, with an optimal pH range of 8 to 11. On the other hand, Cs^+^ ions were equally adsorbed within a wide range of pHs.

The stability of Cs^+^/Pb^2+^ saturated sorbents subjected to the standard leaching tests corresponded well with the quality and effectiveness of adsorption in particular; the better parameters of Pb^2+^ adsorption, following the Langmuir adsorption model, indicated a high binding energy of surface complexation that resulted in a considerably higher stability of WBD that was saturated with Pb^2+^ when compared to Cs^+^. The stability of Pb^2+^ saturated WBDs was about three orders in magnitude higher than that of Cs^+^. In all aspects, LB proved to be a much better sorbent than TN. Both WBDs (LB and TN) are promising prospective adsorbents for Pb^II^.

## Figures and Tables

**Figure 1 materials-12-01647-f001:**
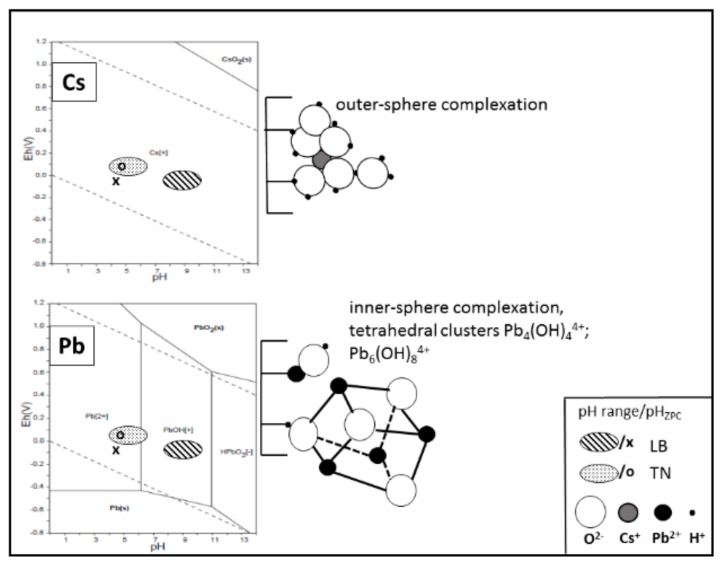
Surface chemistry and expected complexation of Cs^+^/Pb^2+^ in aqueous system.

**Figure 2 materials-12-01647-f002:**
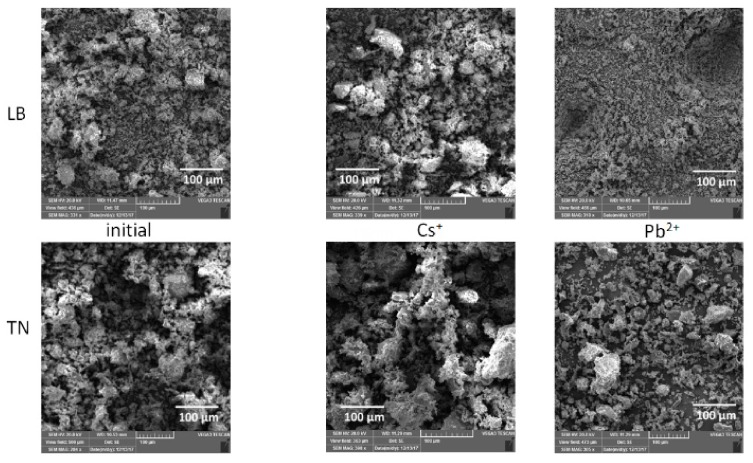
SEM images of WBD before and after Cs^+^/Pb^2+^adsorption on LB and TN (magnification of 300×).

**Figure 3 materials-12-01647-f003:**
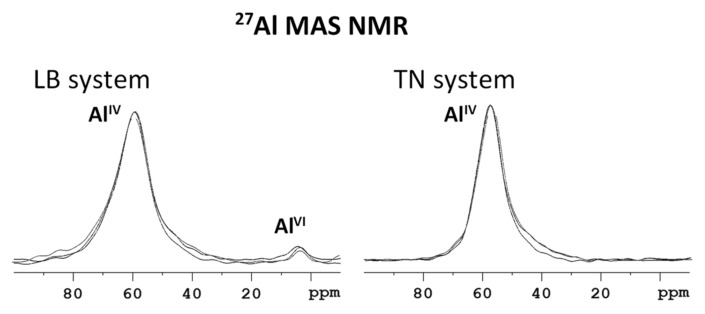
Comparison of ^27^Al MAS NMR spectra of neat LB and TN WBD systems, before and after Cs^+^/Pb^2+^ adsorption.

**Figure 4 materials-12-01647-f004:**
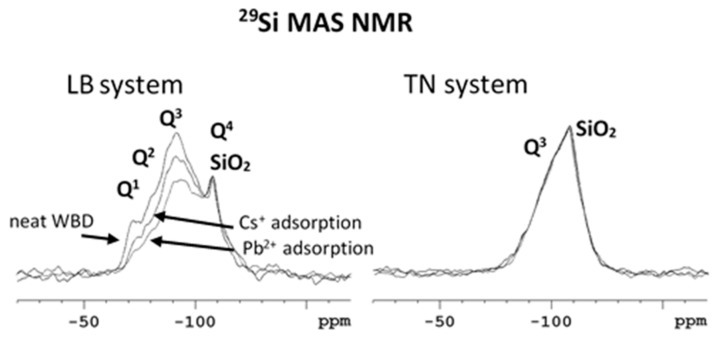
Comparison of ^29^Si MAS NMR spectra of neat LB and TN WBD systems, before and after Cs^+^/Pb^2+^ adsorption.

**Figure 5 materials-12-01647-f005:**
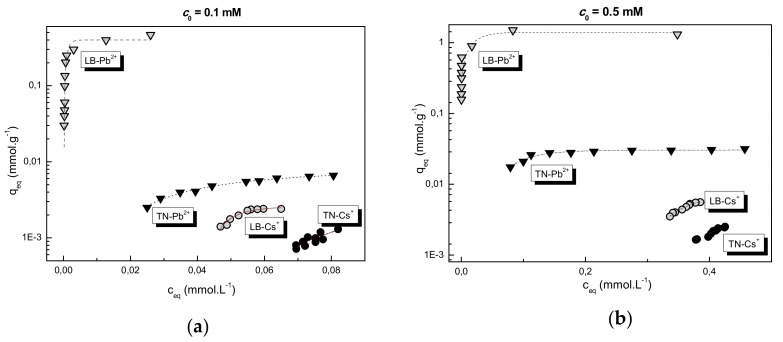
Langmuir adsorption isotherms of Pb^2+^/Cs^+^ adsorption on LB and TN at different initial concentrations; (**a**) *c*_0_ = 0.1 mmol L^−1^, (**b**) *c*_0_ = 0.5 mmol L^−1^. Sorbent dosage ≈ 1–15 g L^−1^ for Pb^2+^ and 1–30 g L^−1^ for Cs^+^.

**Figure 6 materials-12-01647-f006:**
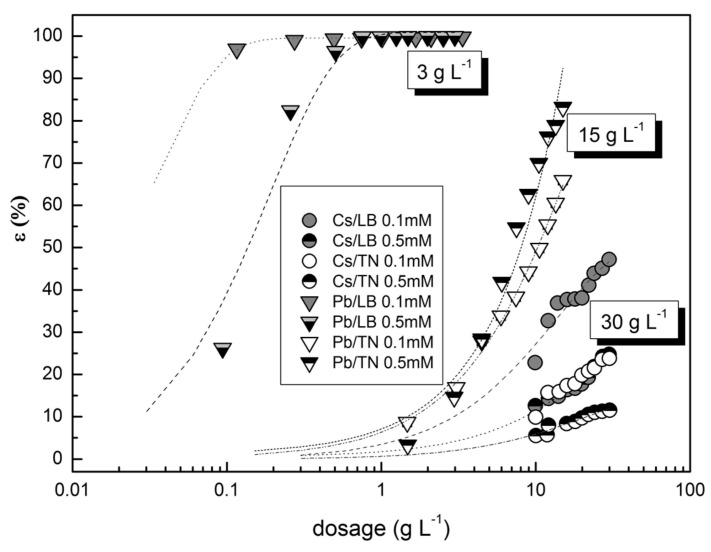
Adsorption efficiency of Cs^+^/Pb^2+^ adsorption on LB and TN in relation to sorbent dosage.

**Figure 7 materials-12-01647-f007:**
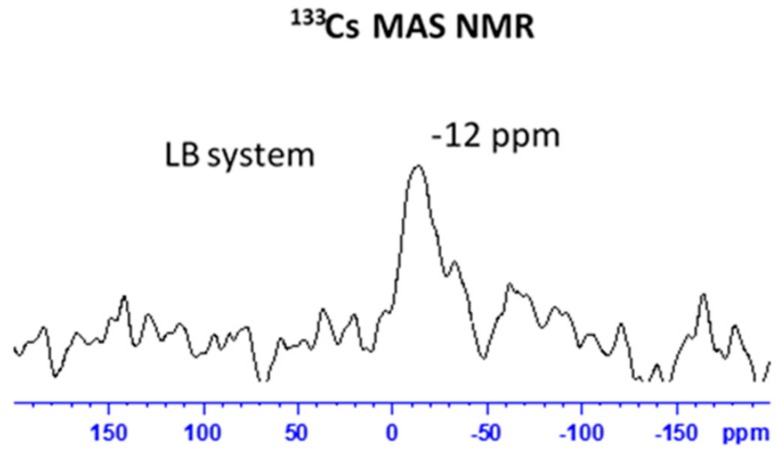
^133^Cs MAS NMR spectrum of LB system after Cs^+^ adsorption.

**Figure 8 materials-12-01647-f008:**
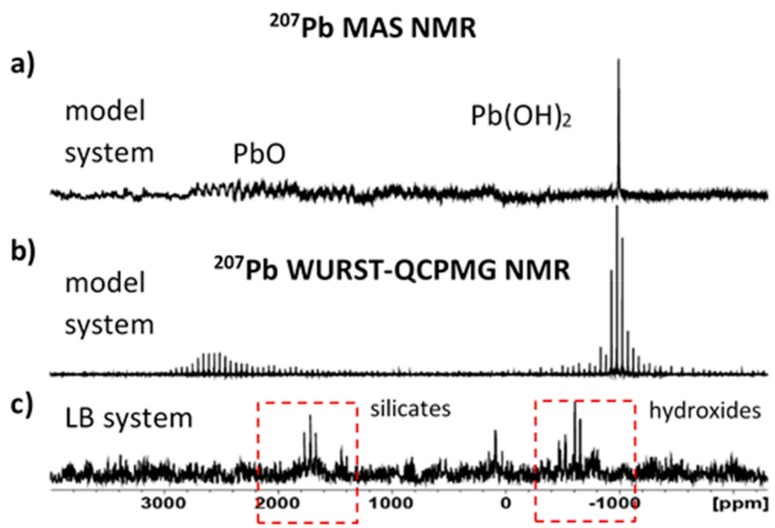
^207^Pb solid-state NMR spectra. (**a**) Standard 207 Pb MAS NMR spectrum of a reference mixture of neat powdered PbO and Pb(OH)_2_ recorded at 12 kHz (NS = 50,000); (**b**) the ^207^Pb WURST-QCPMG NMR spectrum of the reference system (NS = 8000); and (**c**) the corresponding ^207^Pb WURST-QCPMG NMR spectrum of LB system after Pb^2+^ adsorption (NS = 50,000).

**Table 1 materials-12-01647-t001:** Chemical composition, pH of the zero point of charge (pH_ZPC_) and specific surface area (S_BET_) of waste brick dust (WBD) from Libochovice (LB) and Tyn nad Vltavou (TN).

Composition (wt%)	LB	TN
Na_2_O/K_2_O	1.1/4.3	2.5/2.8
MgO/CaO	1.5/14.4	1.3/1.6
Al_2_O_3_	19.7	32.6
SiO_2_	46.4	60.3
P_2_O_5_	0.4	0.3
SO_3_	1.5	0.6
Fe_2_O_3_	3.7	5.4
TiO_2_	0.4	0.8
MnO_2_	0.03	0.04
S_BET_ (m^2^ g^−1^)	4.1	2.5
V_total_ (cm^3^ g^−1^)^*)^	0.01	0.006
pH ^**)^	~9–10	~5–6
pH_ZPC_	4.3	5.5

*) total pore volume of mesopores; **) water leachate at 20 °C, solid-liquid ratio 1:30

**Table 2 materials-12-01647-t002:** Ionic radius and thermodynamic properties of hydrated Cs^+^/Pb^2+^ [25].

Cation	r(Å)	r′(Å)	n	ΔG_hydr_ (kJ mol^−1^)
Cs^+^	1.69	3.28	2.1	−250
Pb^2+^	1.32	4.03	6.1	−1425

r—ionic radius; r′—hydrated radius; n—number of water molecules in hydration shell; ΔG_hydr_—molar Gibbs energy of ion hydration.

**Table 3 materials-12-01647-t003:** Adsorption parameters for Pb and Cs adsorption from model solutions on LB and TN.

Sorbent/c_0_ (mM)	Pb^2+^				Cs^+^			
	q_max_ (mmol g^−1^)	Q_theor_ (mmol g^−1^)	K_L_ (L mmol^−1^)	R^2^	q_max_ (mmol g^−1^)	Q_theor_ (mmol g^−1^)	K_L_ (L mmol^−1^)	R^2^
LB/0.5	1.32	1.6	776.4	0.973	5.5 × 10^−3^	-	-	-
LB/0.1	0.85	1.1	171.5	0.930	2.4 × 10^−3^	-	-	-
TN/0.5	3.4 × 10^−2^	4.5 × 10^−2^	16.7	0.827	2.9 × 10^−3^	-	-	-
TN/0.1	5.5 × 10^−3^	6.8 × 10^−3^	39.8	0.838	1.1 × 10^−3^	-	-	-

**Table 4 materials-12-01647-t004:** Leachability of Cs/Pb from WBDs.

Element/Initial Concentration	Initial Amount in Saturated WBD (mg·g^−1^)	Leached in H_2_O (%)	Class of Leachability*
	Libochovice (LB)		
Cs^+^/0.1 mM	0.32	3.7	not limited
Cs^+^/0.5 mM	0.74	7.4	not limited
Pb^2+^/0.1 mM	146.3	1.2 × 10^−3^	IIa—other waste
Pb^2+^/0.5 mM	212.8	6.6 × 10^−4^	I—inert waste
	Tyn nad Vltavou (TN)		
Cs^+^/0.1 mM	0.15	12.3	not limited
Cs^+^/0.5 mM	0.33	11.4	not limited
Pb^2+^/0.1 mM	0.93	0.25	IIa—other waste
Pb^2+^/0.5 mM	5.98	0.18	IIa—other waste

* Regulation No. 294/2005 (CSN EN 12457).

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
