# Peer review of "Waste Brick Dust as Potential Sorbent of Lead and Cesium from Contaminated Water"

_materials, 2019, doi:10.3390/ma12101647_

Round 1
Reviewer 1 Report
Manuscript entitled “Waste brick dust as potential sorbent of lead and cesium from contaminated water” submitted by Barbora Doušová, David Koloušek, Miloslav Lhotka, Martin Keppert, Martina Urbanová, Libor Kobera and Jiří Brus, can be accepted for publishing in the Materials Journal, after a major revision.
In this study, adsorption properties of waste brick dust (WBD) were studied for the removal of Pb(II) and Cs(I) from aqueous media. The manuscript presents original results, which are relatively organized and systematized. Unfortunately, the interpretation of experimental results is sometimes poor, and this should be corrected before that the manuscript is accepted for publication.
Here is a list of my specific comments:
Page 2, line 52: “…new natural and synthetic adsorbents, preferably low-cost”. Add here as reference the paper: Valorisation possibilities of exhausted biosorbents loaded with metal ions – A review, Journal of Environmental Management, 224 (2018), 288-297, because it is relevant for this observation.
Page 3, lines 104-105: “All adsorption data were fitted to the Langmuir equation [17,18], which was verified as a suitable adsorption model for natural oxides…” How was done this verification??? Also, the mathematical equation of Langmuir isotherm model must be added here.
Page 4, line 134: Why was necessary “Solid-state NMR spectra”??? Why not FTIR spectra. These spectra are more useful.
Page 5, 3.1 Characterization of WBD adsorbents in relation to Pb/Cs chemistry: (a) The effect of pH should be discussed separately. (b) At the end of this section, the optimal pH value selected for the adsorption experiments should be mentioned.
Page 6, lines 193-198: “The SEM micro-observation of initial and Cs+ /Pb2+ 193 saturated WBDs…” (a) This discussion is pertinent only is certain experimental conditions. So these conditions must be mentioned. (b) SEM images cannot highlight the presence of metal ions on surface. So pay attention on these explanations.
Fig. 3 and Fig. 4: As is expected, NMR spectra does not offer too much information about the adsorption of metal ions. May be is better to be replaced by FTIR spectra.
Page 7, 3.3 Adsorption of Pb2+/Cs+ on WBDs: In this section pa attention on the experimental results obtained after adsorption experiments. The using of Langmuir isotherm model provides very useful information’s and quantitative parameters which must be mentioned and discussed here.
Author Response
Response to Reviewer 1 Comments
Point 1: Page 2, line 52: “…new natural and synthetic adsorbents, preferably low-cost”. Add here as reference the paper: Valorisation possibilities of exhausted biosorbents loaded with metal ions – A review, Journal of Environmental Management, 224 (2018), 288-297, because it is relevant for this observation.
Response 1: We accepted the recommendation and added above mentioned manuscript to the references, in place of ref. 12.
Point 2: Page 3, lines 104-105: “All adsorption data were fitted to the Langmuir equation [17,18], which was verified as a suitable adsorption model for natural oxides…” How was done this verification??? Also, the mathematical equation of Langmuir isotherm model must be added here.
Response 2: This paragraph was reformulated and supplemented with some references concerning the use of Langmuir fit for the adsorption on clays, aluminosilicates and metal oxides. We also added the mathematical expressions of Langmuir equations to clarify this part of experimental section.
Point 3: Page 4, line 134: Why was necessary “Solid-state NMR spectra”??? Why not FTIR spectra. These spectra are more useful.
Response 3: We definitely agree with the reviewer, that FTIR spectra are useful for the description of adsorbate-adsorbent binding, the arise of new phases, etc. However, sometimes, especially at a low concentration of adsorbed ions, the results may not be unambiguous. Thanks to the perfect NMR team, solid-state NMR spectra give us information about the deep structure of our materials, including small and/or less noticeable changes and local framework defects. Moreover, 133Cs and 207Pb solid-state NMR spectra allowed to reveal the chemical nature and binding character of even very weakly bound Cs+.
Point 4: Page 5, 3.1 Characterization of WBD adsorbents in relation to Pb/Cs chemistry: (a) The effect of pH should be discussed separately. (b) At the end of this section, the optimal pH value selected for the adsorption experiments should be mentioned.
Response 4: We accepted this point and tried to explain the effect of pH and pH/pHZPC, resp. more clearly and separately from next characterisations. The optimal pH range for effective adsorption was also added.
Point 5: Page 6, lines 193-198: “The SEM micro-observation of initial and Cs+ /Pb2+ saturated WBDs…” (a) This discussion is pertinent only is certain experimental conditions. So these conditions must be mentioned. (b) SEM images cannot highlight the presence of metal ions on surface. So pay attention on these explanations.
Response 5: The SEM images did not primarily make for the identification of metals on the sorbent surface, but for the illustration of surface changes resulted from the binding particles, which composition was more or less known. According to the recommendation of Reviewer 1, we specified the conditions in Fig. 2.
Point 6: Fig. 3 and Fig. 4: As is expected, NMR spectra does not offer too much information about the adsorption of metal ions. May be is better to be replaced by FTIR spectra.
Response 6: According to the recommendation of Reviewer 1, we tried to compare NMR and FTIR spectra of WBD adsorbents. Since the Cs/Pb binding was poorly detectable in FTIR spectra thanks to arelatively low concentration, the NMR spectra were finally used for supporting the explanation of adsorption processes via structural changes and defects observed in materials before and after adsorption. This paragraph was reformulated and shortened for a better clarity.
Point 7: Page 7, 3.3 Adsorption of Pb2+/Cs+ on WBDs: In this section pa attention on the experimental results obtained after adsorption experiments. The using of Langmuir isotherm model provides very useful information’s and quantitative parameters which must be mentioned and discussed here.
Response 7: We accepted the recommendation and the Section 3.3 was rewritten, with emphasizing the difference between the two adsorbed cations and discussing that in more details.

Reviewer 2 Report
The study is interesting. However, the discussion is superficial.
Even for the application focused work, a discussion taking only divalent Pb and monovalent Cs seems incomplete.
It seems the brick dust is good for divalent cations. So, it makes sense to compare among the divalent/trivalent heavy metal ions.
Also, it is good to add at least the adsorbent/solution ratio in the figure caption.
Author Response
Response to Reviewer 2 Comments
Point 1: The study is interesting. However, the discussion is superficial.
Response 1: Although this recommendation has not been specified, we tried to improve the discussion in several parts: (i) Langmuir adsorption model was described separately in the Experimental part; (ii) the discussion concerning adsorption of cations was rewritten and enlarged; (iii) effect of pH and pHZPC on the adsorption selectivity and running was explained more clearly (as we hope) and in more detail.
Point 2: Even for the application focused work, a discussion taking only divalent Pb and monovalent Cs seems incomplete.
Response 2: The applicability of WBD in adsorption processes was previously studied by Dousova et al. (2016), where the adsorption of ecologically harmful cations (Cd, Pb, Cs) and anions (As, Sb, Cr, U) on WBD were tested and evaluated. In this paper Pb2+ and Cs+ were selected as very toxic on the one hand and suitable for the study of structural and binding properties on the other hand. According to the recommendation of Reviewer 2 we enlarged the explanation of previous work and empasized the connection of the both studies.
Point 3: It seems the brick dust is good for divalent cations. So, it makes sense to compare among the divalent/trivalent heavy metal ions.
Response 3: This is a very motivating idea – heavy metals forming trivalent cations are more interesting as rare elements than toxic ones, therefore, they possible collection and/or separation with the use of WBD calls for a detailed individual study.
Point 4: Also, it is good to add at least the adsorbent/solution ratio in the figure caption.
Response 4: We thanks the Reviewer 2 for this notification, the sorbent dosage was added to the caption of fig. 3.
Round 2
Reviewer 1 Report
All my previous remarks and comments have been considered in this new version of the manuscript. From my point of view, the revised manuscript meets the criteria and can be published as original paper in Materials Journal.
Reviewer 2 Report
The manuscript is improved well as most of the sections of concern are addressed,